

# Atorvastatin alters the expression of genes related to bile acid metabolism and circadian clock in livers of mice

Wen-Kai Li[1,2], Huan Li[1], Yuan-Fu Lu[1], Ying-Ying Li[1], Zidong Donna Fu[3] and Jie Liu[1]

[1] Key Lab for Basic Pharmacology of Ministry of Education and Joint International Research Laboratory of Ethnomedicine of Ministry of Education, Zunyi Medical College, Zunyi, China
[2] Department of Pharmacology, Shanghai University of Chinese Traditional Medicine, Shanghai, China
[3] Department of Environmental and Occupational Health Sciences, University of Washington, Seattle, WA, United States of America

## ABSTRACT

**Aim**. Atorvastatin is a HMG-CoA reductase inhibitor used for hyperlipidemia. Atorvastatin is generally safe but may induce cholestasis. The present study aimed to examine the effects of atorvastatin on hepatic gene expression related to bile acid metabolism and homeostasis, as well as the expression of circadian clock genes in livers of mice.

**Methods**. Adult male mice were given atorvastatin (10, 30, and 100 mg/kg, po) daily for 30 days, and blood biochemistry, histopathology, and gene expression were examined.

**Results**. Repeated administration of atorvastatin did not affect animal body weight gain or liver weights. Serum enzyme activities were in the normal range. Histologically, the high dose of atorvastatin produced scattered swollen hepatocytes, foci of feathery-like degeneration, together with increased expression of Egr-1 and metallothionein-1. Atorvastatin increased the expression of Cyp7a1 in the liver, along with FXR and SHP. In contract, atorvastatin decreased the expression of bile acid transporters Ntcp, Bsep, Ost$\alpha$, and Ost$\beta$. The most dramatic change was the 30-fold induction of Cyp7a1. Because Cyp7a1 is a circadian clock-controlled gene, we further examined the effect of atorvastatin on clock gene expression. Atorvastatin increased the expression of clock core master genes Bmal1 and Npas2, decreased the expression of clock feedback genes Per2, Per3, and the clock targeted genes Dbp and Tef, whereas it had no effect on Cry1 and Nr1d1 expression.

**Conclusion**. Repeated administration of atorvastatin affects bile acid metabolism and markedly increases the expression of the bile acid synthesis rate-limiting enzyme gene Cyp7a1, together with alterations in the expression of circadian clock genes.

# INTRODUCTION

Statins are the primary drugs clinically used to treat hyperlipidemia (*Grover, Luthra & Maroo, 2014*; *Kalantari & Naghipour, 2014*). As the number of hypercholesterolemic

Corresponding author
Jie Liu, Jie@liuonline.com

patients increases, statins, which are 3-hydroxy-3-methylglutaryl coenzyme A (HMG-CoA) reductase inhibitors, become more important. Statins are generally safe in clinical use (*Kalantari & Naghipour, 2014*), but there are few adverse drug reaction reports that are associated with statins, such as hepatotoxicity (*Bjornsson, Jacobsen & Kalaitzakis, 2012*; *Bjornsson, 2015*; *Grover, Luthra & Maroo, 2014*; *Kalantari & Naghipour, 2014*).

Cholestasis is one of major adverse effects in statin-induced liver injury. It has been reported that atorvastatin induces prolonged cholestasis in patients (*Beltowski, Wojcicka & Jamroz-Wisniewska, 2009*; *Merli et al., 2010*). Accumulated bile acids could act as inflammagens, and directly activate signaling pathways in hepatocytes that stimulate production of proinflammatory mediators (*Allen, Jaeschke & Copple, 2011*). It has been shown that the early growth response protein-1 (Egr-1) signal pathway and metallothionein-1 (MT-1) are involved in acute intrahepatic cholesteric liver injury (*Allen, Jaeschke & Copple, 2011*; *Alscher et al., 2002*; *Ding et al., 2008*; *Sullivan et al., 2012*).

Cholestasis is related to disruption of bile acid (BA) homeostasis (*Parker et al., 2013*). Hydrophobic BAs are potent inflammatory chemical that cause injury to liver (*Chiang, 2013*). In vitro and *in vivo* studies indicate that statins can alter BA metabolism in hepatocytes (*Byun et al., 2014*; *Fu, Cui & Klaassen, 2014*). The rate-limiting enzyme for the classic pathway of BA synthesis in the liver is Cyp7a1(*Chiang, 2013*). It has been reported that atorvastatin increases Cyp7a1, and alters the expression of Cyp7a1 regulatory genes, farnesoid X receptor (FXR) and small heterodimer partner (SHP) (*Byun et al., 2014*; *Fu, Cui & Klaassen, 2014*). BA transporters also play pivotal roles in maintaining BA homeostasis. BAs are excreted into the bile by the bile salt export pump (Bsep), and transported from the liver into blood by multidrug resistance-associated protein 3 and 4 (Mrp3 and Mrp4), and organic solute transporter $\alpha$ and $\beta$ dimer (Ost$\alpha$/$\beta$) (*Chiang, 2013*; *Fu, Cui & Klaassen, 2014*). The organic anion transporting polypeptide (Oatp1b2) and $Na^+$-dependent taurocholate cotransport peptide (Ntcp) are responsible for BA uptake into hepatocytes (*Cheng, Buckley & Klaassen, 2007*; *Chiang, 2013*; *Csanaky et al., 2011*). Thus, an understanding of the effects of statins on bile acid homeostasis could be important in predicting the mechanism of statin-induced cholestasis.

The rate-limiting enzyme for bile acid homeostasis is Cyp7a1, which converts cholesterol to 7$\alpha$-hydroxcholesterol (*Chiang, 2013*). Statins have also been shown to influence the expression of Cyp7a1(*Fu, Cui & Klaassen, 2014*; *Jiang et al., 2012*; *Kolouchova et al., 2011*). In one of our recent studies, atovarstatin at 100 mg/kg, po for 7 days, induced Cyp7a1 (*Fu, Cui & Klaassen, 2014*), and whether such effects can be observed at lower doses and longer-times is our primary goals. Cyp7a1 is a circadian clock-driven gene displaying a typical circadian rhythm (*Kovar et al., 2010*; *Zhang, Guo & Klaassen, 2011*), with a peak/trough ratio over 10 fold (*Lu et al., 2013*). It is known that circadian dysregulation disrupts bile acid homeostasis (*Ferrell & Chiang, 2015b*; *Ma et al., 2009*), and diurnal variation of hepatic cholesterol synthesis is driven primarily by varying the steady-state mRNA levels for HMG-CoA reductase (*Jurevics et al., 2000*). However, little is known about the effects of statins on circadian rhythm, and this study is aimed to fill the gap.

Therefore, the goals of the present study are two-fold: (1) to examine the effect of repeated administration of atorvastatin on bile acid homeostasis and cholestasis-associated

inflammation; and (2) to examine the effects of atorvastatin on the expression of Cyp7a1 and circadian clock genes. The results demonstrate that atorvastatin affects bile acid homeostasis, producing about 30-fold induction of Cyp7a1, and this effect is associated with dysregulation of circadian clock genes.

## MATERIALS AND METHODS

### Animals and treatment

Adult male Kunming mice were obtained from the Experimental Animal Center of the Third Military Medical University (Chongqing, China) and randomly assigned to four groups. Mice were maintained in a room at $22 \pm 2$ °C with a 12 h light-dark cycle, and had free access to standard rodent chow and water. The control group of mice ($n = 4$) received saline and the treated groups ($n = 5$/per group) were given atorvastatin (Topfond Pharmaceutical Co., Ltd, Henan, China) orally at doses of 10, 30 and 100 mg/kg once daily for one month. The dose selection was based on human-mice dose conversion (about 10 mg/kg, po), repeated doses in dog (up to 30 mg/kg, po for 13 weeks) (*Herron et al., 2015*) and the dose (100 mg/kg, po for 7 days) used for Cyp7a1 induction (*Fu, Cui & Klaassen, 2014*). The animal body weights were recorded every three days. In the morning at 10:00 am (1 h after the last dose of atovarstatin), the mice were anesthetized with 7% chloral hydrate, and blood and livers were collected. All animal experiments were carried out in full compliance with the Guidance of Humane Care and Use of Laboratory Animals, and approved by the Animal Care and Use Committee of Zunyi Medical College (2013–5).

### Blood biochemistry

The blood was kept at room temperature for one hour, and serum was separated by centrifugation. Serum levels of alanine aminotransferase (ALT) were measured using a commercial kit (Jiangcheng Co, Nanjing, China).

### Histopathological procedure

Liver tissues were placed in plastic cassettes and immersed in 10% formalin for 48 h. The fixed tissues were processed by standard histology procedures and subjected to hematoxylin and eosin (HE) staining. The degree of hepatocellular damage was evaluated by light microscopy.

### RNA isolation and RT-qPCR

Approximately 50–100 mg of liver tissue was homogenized in 1 ml Trizol (TakaRa Biotechnology, Dalian, China)-the quality and quantity of RNA were determined by the 260/280 ratio (>1.8) and by gel-electrophoresis. Total RNA was reversely transcribed with a High Capacity Reverse Transcriptase Kit (Applied Biosystems, Foster City, CA, USA). The primers were designed with Primer3 software and listed in Table 1. The 15 µl PCR reaction mix contained 3 µl of cDNA (10 ng/µl), 7.5 µl of iQ$^{TM}$ SYBR GreenSupermix (Bio-Rad Laboratories, Hercules, CA), 0.5 µl of primer mix (10 µM each), and 4 µl of ddH$_2$O. After 5 min denature at 95 °C, 40 cycles will be performed: annealing and extension at 60 °C for 45 s and denature at 95 °C for 10 s. Dissociation curve was performed after finishing 40

**Table 1 Primer sequence for real-time RT-PCR analysis.**

| Gene | GenBank number | Forward (5′–3′) | Reverse (5′–3′) |
|------|----------------|-----------------|-----------------|
| β-actin | M12481 | GGCCAACCGTGAAAAGATGA | CAGCCTGGATGGCTACGTACA |
| Bmal1 | NM_007489 | ACGACATAGGACACCTCGCAGA | CGGGTTCATGAAACTGAACCATC |
| Bsep | NM_021022 | GGACAATGATGTGCTTGTGG | CACACAAAGCCCCTACCAGT |
| Cry1 | NM_007771 | GGATCCACCATTTAGCCAGACAC | CATTTATGCTCCAATCTGCATCAAG |
| Cyp7a1 | NM_007824 | ATCCTGGCAAACAGAAATCG | GGCCAAGTCTGGTTTCTCTG |
| Dbp | NM_016974 | ATCTCGCCCTGTCAAGCATTC | TGTACCTCCGGCTCCAGTACTTC |
| Egr-1 | NM_031332 | GTCTCTCTCTGGCCTGGTTT | GCACAAAGATGAGGGCCAAA |
| FXR | NM_009108 | TGGGTACCAGGGAGAGACTG | GTGAGCGCGTTGTAGTGGTA |
| MT-1 | NM_013602 | CTCCGTAGCTCCAGCTTCAC | AGGAGCAGCAGCTCTTCTTG |
| Nr1d1 | NM_145434 | GTGAAGACATGACGACCCTGGA | TGCCATTGGAGCTGTCACTGTAG |
| Npas2 | MA151656 | TGCTCCGAGAATCGAATGTGATA | ATGGCAGGCTGCTCAGTGAA |
| Ntcp | NM_011387 | GGTGCCCTACAAAGGCATTA | ACAGCCACAGAGAGGGAGAA |
| Ost α | NM_145932 | TTGTGATCAACCGCATTTGT | TTGTGATCAACCGCATTTGT |
| Ost β | NM_178933 | ATCCTGGCAAACAGAAATCG | GGCCAAGTCTGGTTTCTCTG |
| Per2 | NM_011066 | CCTACAGCATGGAGCAGGTTGA | TTCCCAGAAACCAGGGACACA |
| Per3 | MA164628 | CTCAAGACGTGAGGGCGTTCTA | GGTTTCGCTGGTGCACATTC |
| SHP | NM_011850 | CTCATGGCCTCTACCCTCAA | GGTCACCTCAGCAAAAGCAT |
| Tef | MA032354 | CTTCAACCCTCGGAAGCACA | CCGGATGGTGATCTGGTTCTC |

cycles to verify the quality of primers and amplification. Relative expression of genes was calculated by the $2^{-\Delta\Delta Ct}$-method and normalized to the housekeeping gene β-actin.

## Statistical analysis

Data were expressed as mean and standard error. The SPSS 16 software was used for statistical analysis. Data were analyzed using a one-way analysis of variance (ANOVA), followed by Duncan's multiple range test. $p$ value <0.05 was considered statistically significant.

## RESULTS

### Animal body weight, liver index and serum ALT levels

Repeated administration of atorvastatin did not affect animal body weight gain, and did not alter animal activity and general health. The animal body weight gain in the group treated with 100 mg/kg atorvastatin was slightly lower than other groups, but was not significantly different. The liver weight relative to the body weight ratios (liver index) also showed no significant differences among the four groups (48 ± 2.8, 48 ± 4.2, 51 ± 5.0, and 49 ± 2.1mg liver/g body weight for Control, 10, 30, and 100 mg/kg atorvastatin groups, respectively). Serum ALT is a biomarker of liver injury. The serum levels of ALT in atorvastatin-treated mice were in general not elevated. Serum ALT levels in mice treated with 100 mg/kg atorvastatin were slightly higher than the control group, but were not statistically significant and within the normal range (48 ± 8.2, 48 ± 14, 33 ± 5.4, and 77 ± 14 U/L for Control, 10, 30, and 100 mg/kg atorvastatin groups, respectively).

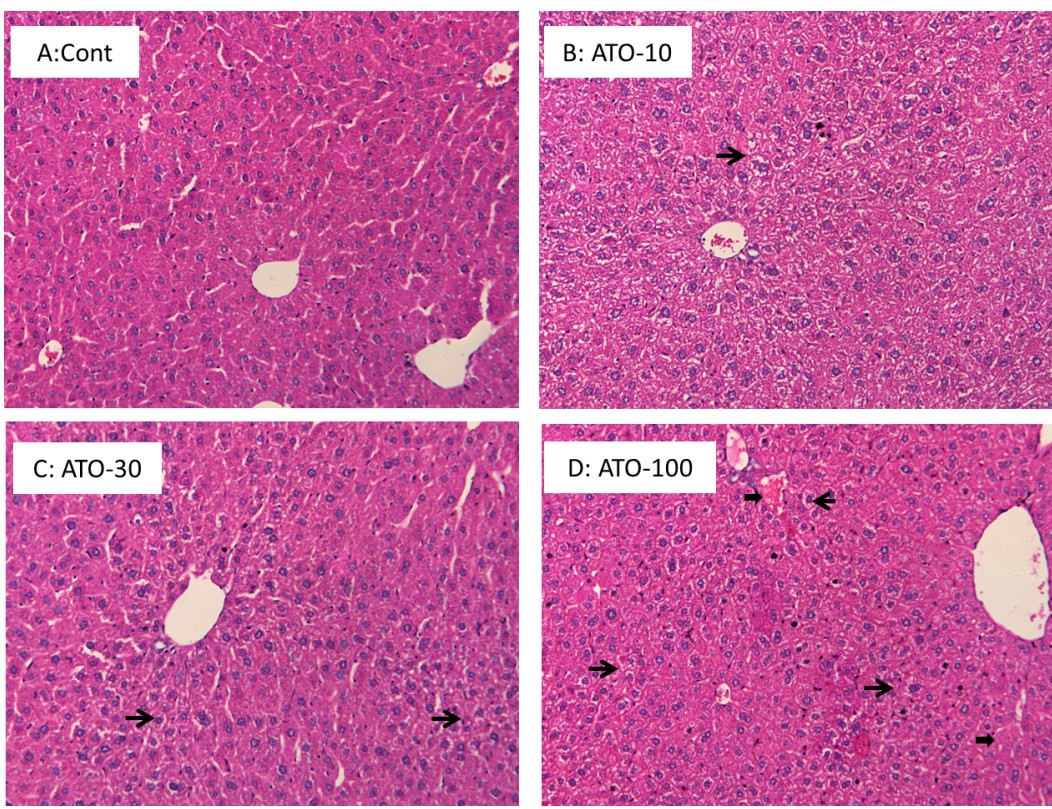

**Figure 1** **Histopathology.** Mice received atorvastatin 10, 30 or 100 mg/kg, po for 30 days. Representative H&E staining of liver sections showed spotted feathery-like degeneration (thin arrows), and spotted blood lakes (peliosishepatis) (thick arrows). Magnitude 200×.

## Histopathological findings

Liver sections from mice administered the various dosages of atorvastatin were stained with H&E and representative photographs are shown in Fig. 1. There were no significant differences between control and atorvastatin groups; however the foci of feathery-like degeneration, indicative of mild cholestasis (thin arrows) (*Li & Crawford, 2004*), were observed. In mice treated with 100 mg/kg atorvastatin, spotted blood lakes (peliosishepatis) can be seen (thick arrows).

## The expression of Egr-1 and MT-1 in liver

Cholestasis contributes to liver injury and is associated with increased expression of numerous proinflammatory mediators (*Allen, Jaeschke & Copple, 2011*). In the present study, atorvastatin significantly increased the expression of Egr-1 and MT-1, indicative of an inflammatory response, in the livers of mice treated with atorvastatin for 30 days (Fig. 2).

## The expression of genes related to BA-homeostasis

Cyp7a1 encodes the rate-limiting enzyme for the classic pathway of BA synthesis in liver (*Chiang, 2013*). The mRNA of Cyp7a1 in the atorvastatin-treated mice was markedly higher than in the control mice (26.1-fold at 10 mg/kg, 33.8-fold at 30 mg/kg, and 14-fold

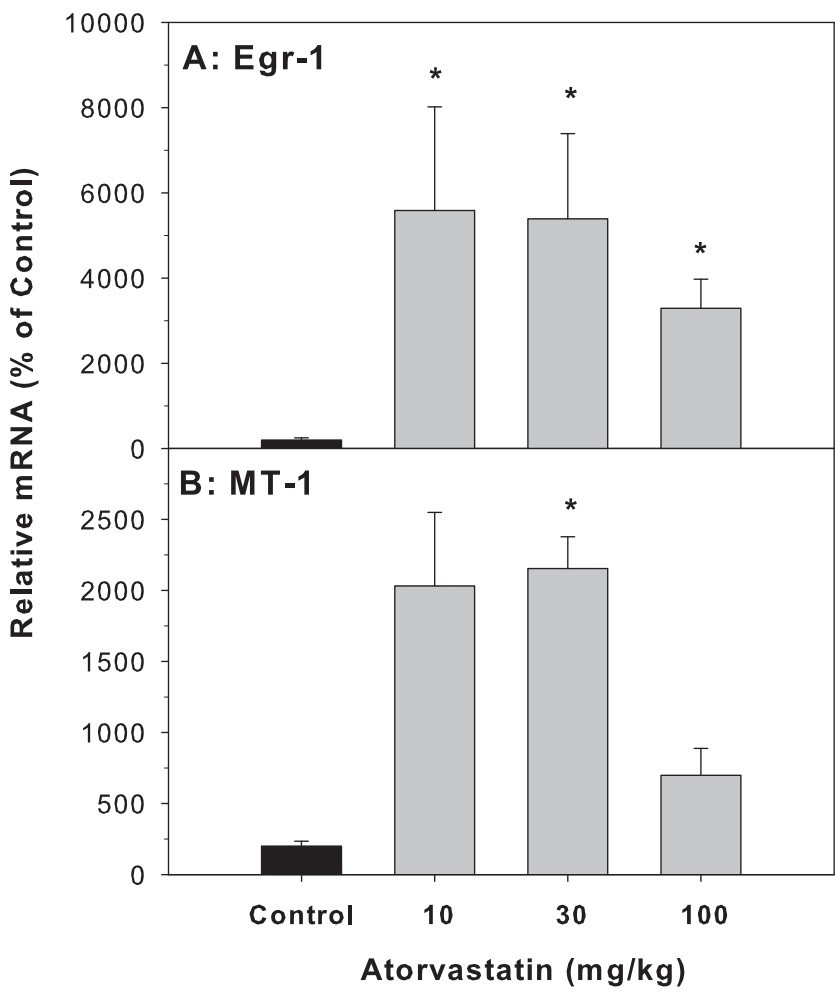

**Figure 2 Effects of atorvastatin treatment on mRNA expression of Egr-1 and MT-1 in mouse livers.** Mice were given atorvastatin 10, 30 or 100 mg/kg, po for 30 days. Total liver RNA was extracted and subjected to RT-PCR analysis. Data represents the mean ± SE of $n = 5$. *Significantly different from control mice, $p < 0.05$.

at 100 mg/kg of atorvastatin). FXR is a bile acid-activated nuclear receptor. FXR induces transcription of the negative nuclear receptor, small heterodimer partner (SHP), and SHP down-regulates Cyp7A1. In the present study, atorvastatin increased the expression of FXR and SHP (Fig. 3).

## Effects of mRNA expression of bile acid transporters

The major transporters of bile acids in the liver include Bsep, Ntcp, Ost-α and Ost-β (*Chiang, 2013*; *Klaassen & Aleksunes, 2010*). The mRNA of Bsep (45.8–78.7% of controls), Ntcp (55.3–70.8% of controls), Ost-α (20.1–36.7% of controls) and Ost-β (28.9–58.8% of controls) was decreased after atorvastatin treatment (Fig. 4).

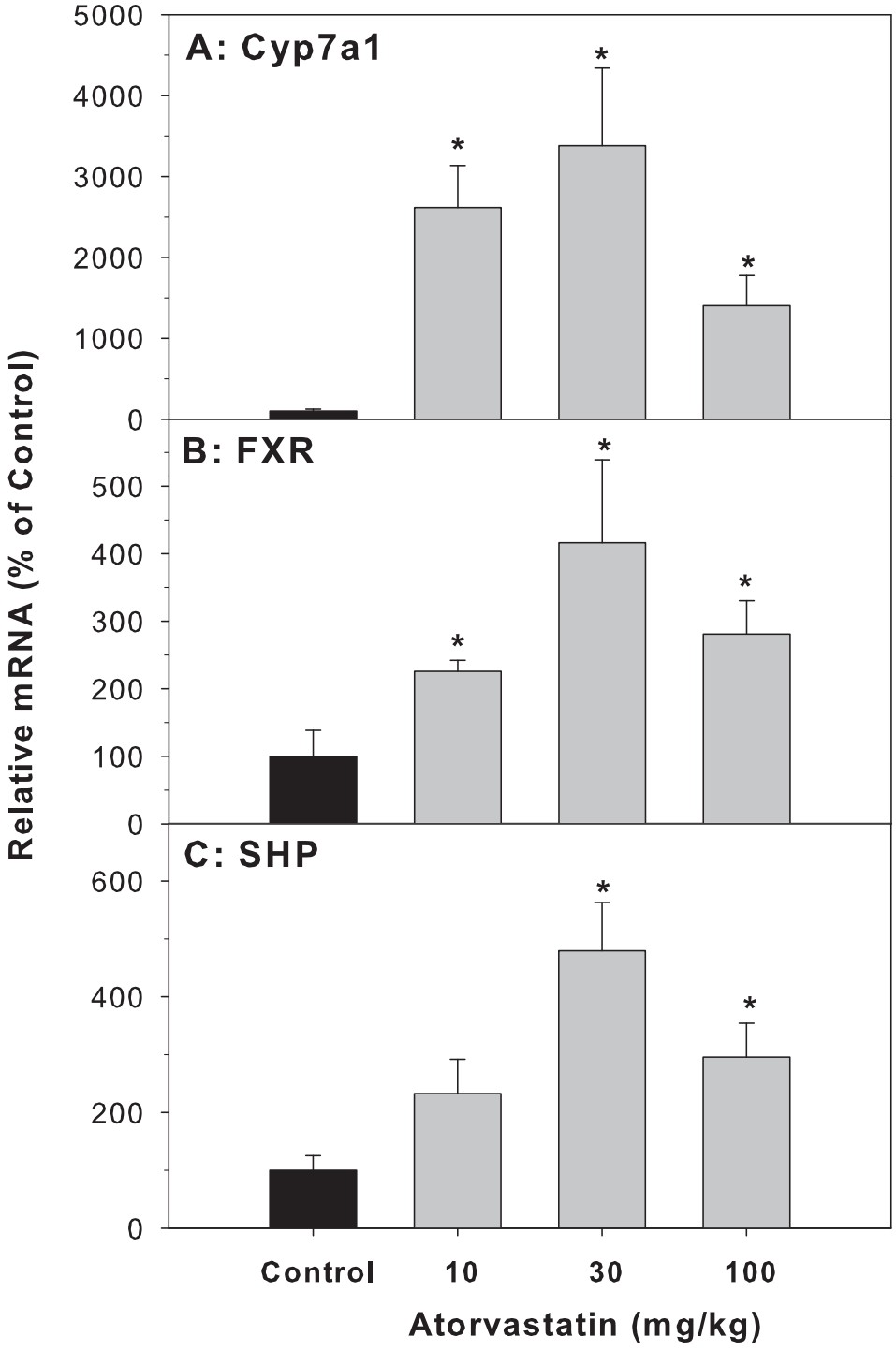

**Figure 3** **Effects of atorvastatin treatment on mRNA expression of genes related to bile acid metabolism.** Mice were given atorvastatin 10, 30 or 100 mg/kg, po for 30 days. Total liver RNA was extracted and subjected to RT-PCR analysis. Data represent the mean ± SE of $n = 5$. *Significantly different from control mice, $p < 0.05$.

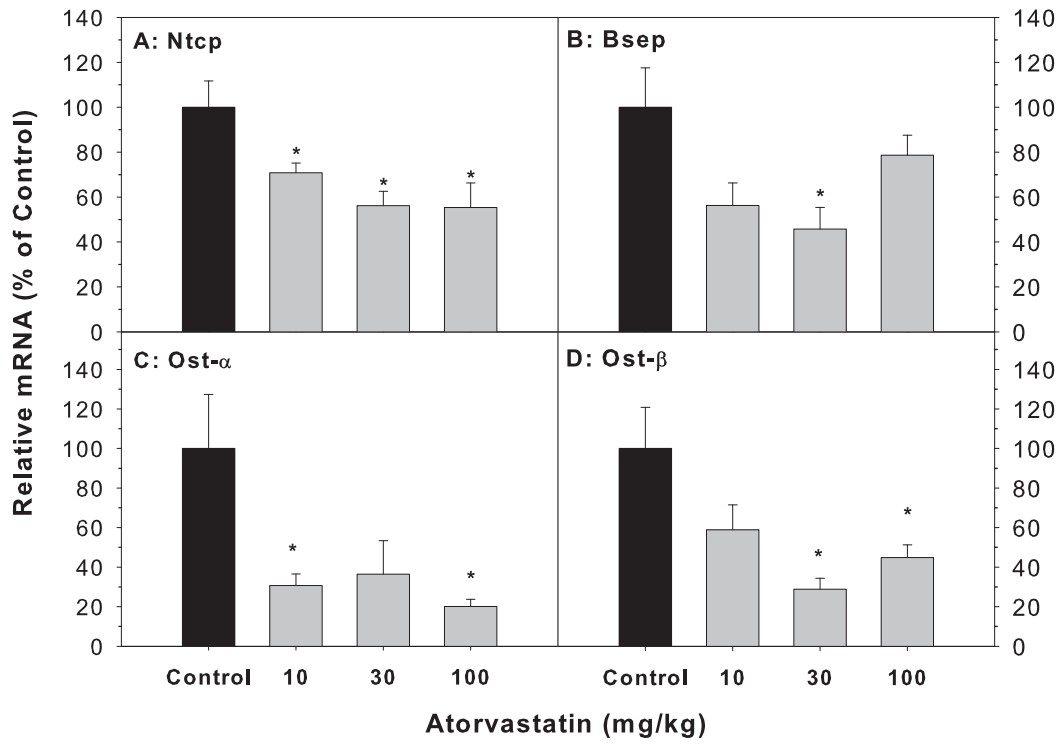

**Figure 4** **Effects of atorvastatin treatment on mRNA expression of bile acid transporters.** Mice were given atorvastatin 10, 30 or 100 mg/kg, po for 30 days. Total liver RNA was extracted and subjected to RT-PCR analysis. Data represent the mean $\pm$ SE of $n = 5$. *Significantly different from control mice, $p < 0.05$.

## Effects of mRNA expression of circadian genes in liver

Because the most dramatic effect of atorvastatin was the 33.8-fold induction of Cyp7a1, which is a clock-controlled gene (*Lu et al., 2013*), we further examined the effect of atorvastatin on the expression of clock genes. The Cyp7A1 gene promoter is under transcriptional control of the clock brain and muscle Arnt-like protein-1 (Bmal1) and other clock genes (*Noshiro et al., 2007*). The mRNA expression of Bmal1 and neuronal PAS domain protein 2 (Npas2) was increased by atorvastatin, in a similar trend as the Cyp7a1 increases (Fig. 5).

For clock feedback control genes, atorvastatin decreased the expression of period circadian protein gene 2 (Per 2) and Per 3, but had no effects on the mRNA expression of cryptochrome 1 (Cry1) (Fig. 6).

For clock-targeted/driven genes, the expression of D site albumin promoter binding protein (Dbp) and thyrotroph embryonic factor (Tef) in atorvastatin-treated mice was significantly lower than that of the control group, while the expression of nuclear receptor subfamily 1, group D, member 1 (Nr1d1) in atorvastatin-treated mice was unchanged (Fig. 7).

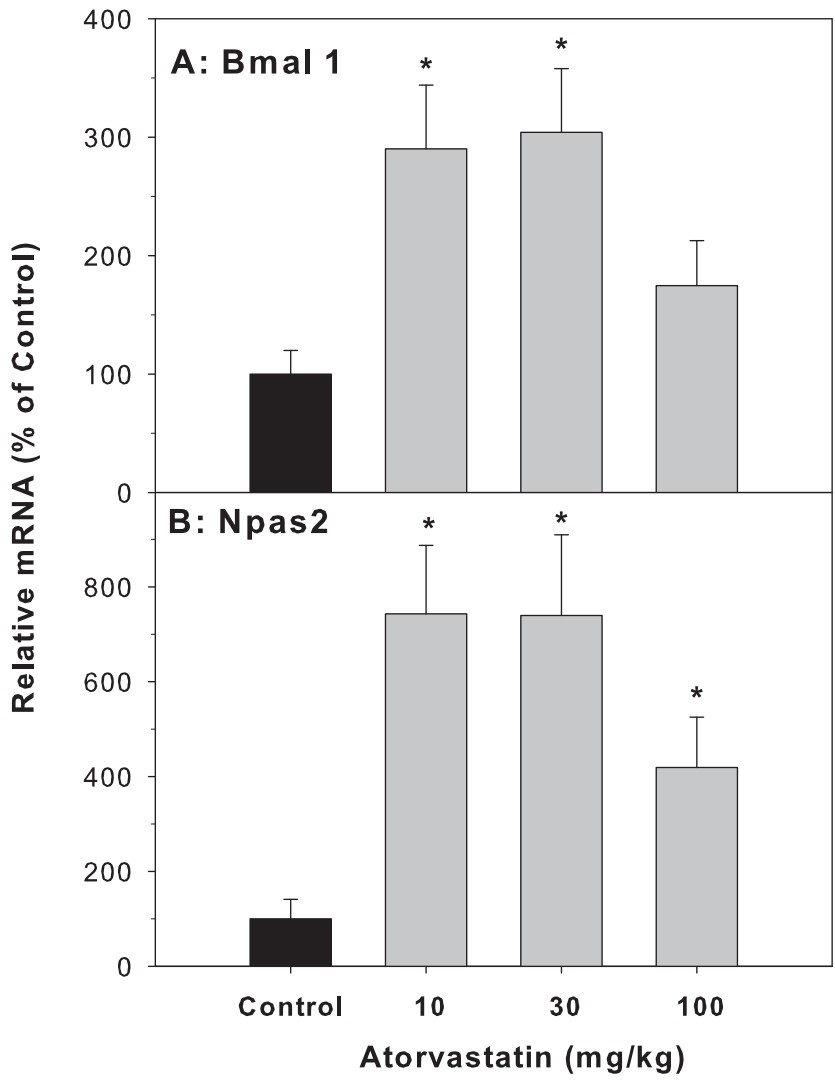

**Figure 5** **Effects of atorvastatin treatment on mRNA expression of clock core master genes Bmal1 and Npas2.** Mice were administered atorvastatin 10, 30 or 100 mg/kg, po for 30 days. Total liver RNA was extracted and subjected to RT-PCR analysis. Data represent the mean ± SE of 4–5 mice. *Significantly different from control mice, $p < 0.05$.

## DISCUSSION

The present study showed that repeated administration of high doses of atorvastatin did not alter animal body weight gain, liver weight, serum ALT, or cause overt pathological alterations except for foci of inflammation, suggesting that atorvastatin is generally safe. However, at these doses atorvastatin increased the gene expression of Cyp7a1 as previously reported (*Fu, Cui & Klaassen, 2014*). Because Cyp7a1 is a clock-controlled gene (*Lu et al., 2013*), we further examined the effects of atorvastatin on the expression of circadian clock genes, and found Atorvastatin increased the expression of clock core master genes Bmal1 and Npas2, decreased the expression of clock feedback genes Per2, Per3, and the clock targeted genes Dbp and Tef in the livers of mice.

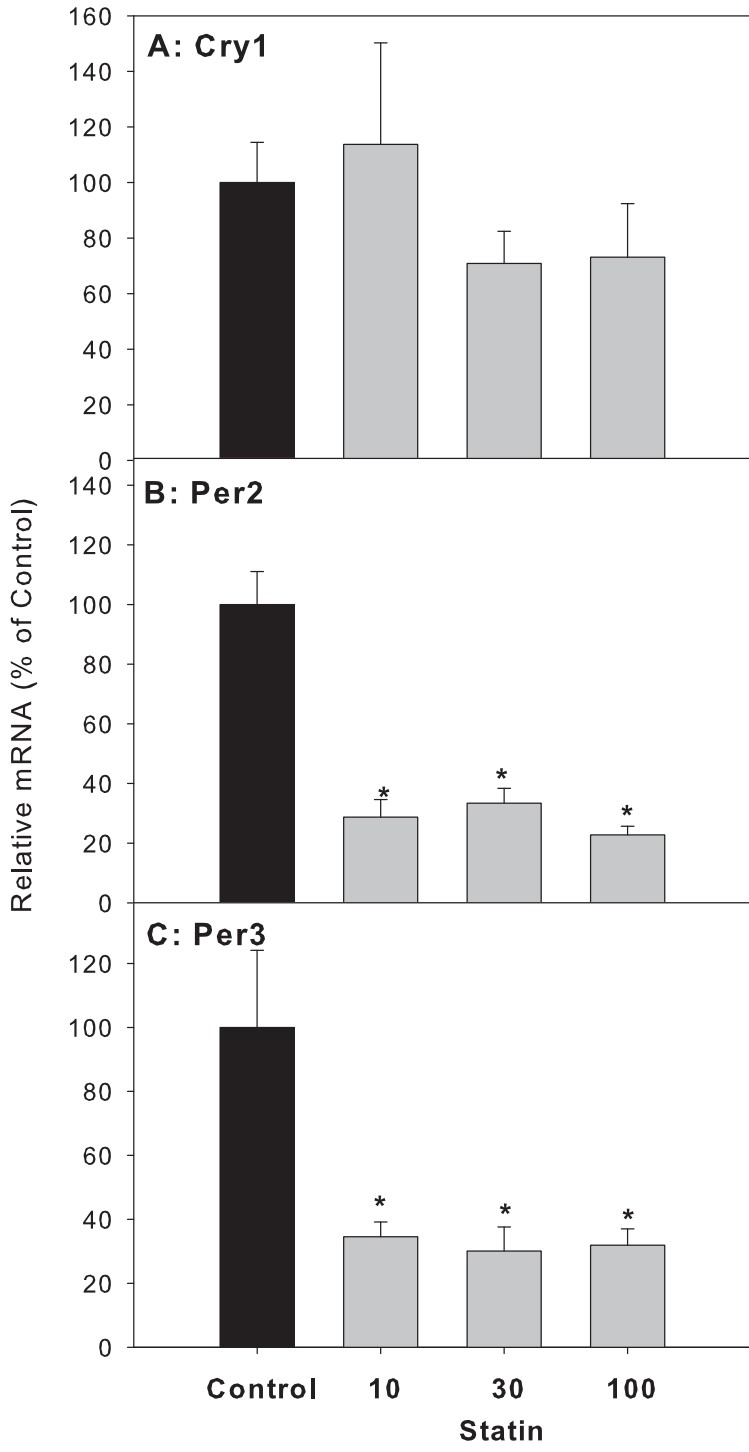

**Figure 6** **Effects of atorvastatin treatment on mRNA expression of clock feedback control genes Per2, Per3 and Cry1.** Mice were given atorvastatin 10, 30 or 100 mg/kg, po for 30 days. Total liver RNA was extracted and subjected to RT-PCR analysis. Data represent the mean $\pm$ SE of $n = 5$. *Significantly different from control mice, $p < 0.05$.

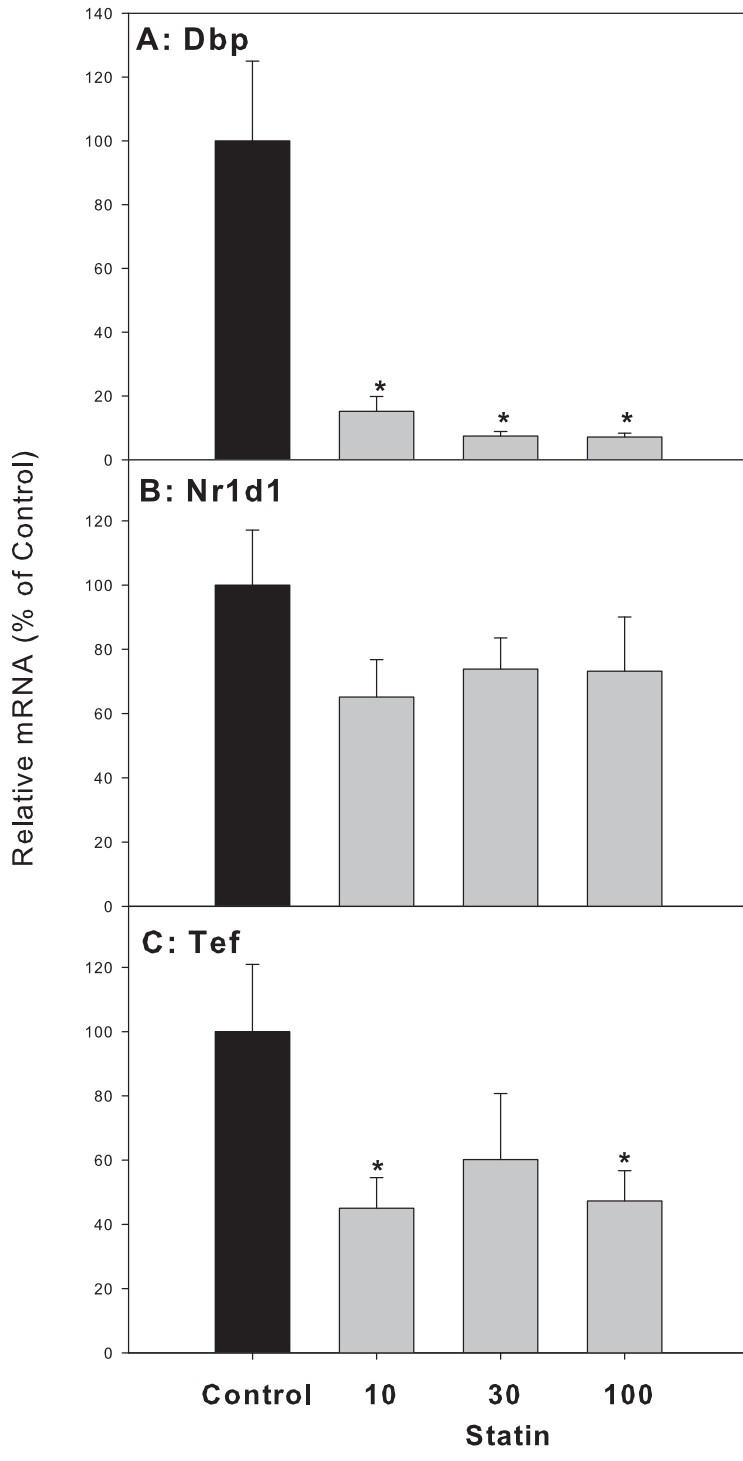

**Figure 7** **Effects of atorvastatin treatment on mRNA expression of clock targeted or driven genes Dbp, Tef, and Nr1d1.** Mice were given atorvastatin 10, 30 or 100 mg/kg, po for 30 days. Total liver RNA was extracted and subjected to RT-PCR analysis. Data represent the mean $\pm$ SE of $n = 5$. *Significantly different from control mice, $p < 0.05$.

Few studies have shown that statins can induce oxidative stress and elevate serum AST and ALT (*Beltowski, Wojcicka & Jamroz-Wisniewska, 2009*; *Kolouchova et al., 2011*). In the present study, no apparent liver toxicity was observed even after a high dose of atorvastatin (100 mg/kg po for 30 days). This is in agreement with the majority of the literature that statins are relatively safe (*Kalantari & Naghipour, 2014*). However, the high dose of atorvastatin had potential to produce cholestasis. Cholestasis is characterized by decreased bile flow resulting in accumulation of bile acids with spotted feathery-like degeneration in hepatocytes (*Liu et al., 2013*; *Lu et al., 2013*), and is also the main feature of statin-induced liver injury (*Kerzner, Irabagon & Berkelhammer, 2013*; *Merli et al., 2010*; *Russo, Scobey & Bonkovsky, 2009*). Overload of bile acids is a physiological stress that leads to inflammatory-associated gene expression, such as MT, a metal-binding, cysteine-rich protein (*Alscher et al., 2002*), as well as Egr-1, a critical regulator of inflammation and cellular stress (*Kalantari & Naghipour, 2014*; *Kim et al., 2006*; *Russo, Scobey & Bonkovsky, 2009*)). Overexpression of inflammatory genes is accompanied by cholestasis. In the present study the expression of Egr1 was significantly increased, and the expression of MT-1 was also increased, indicative of cholestatic stress produced by atorvastatin.

Cholestasis increases the concentrations of bile acids in liver. Hepatic BA homeostasis is tightly controlled by a variety of regulators within a relatively narrow range (*Chiang, 2013*; *Perez & Briz, 2009*). It is known that both BA synthetic enzymes and BA transporters play an important role in BA homeostasis. The data of the present study showed that atorvastatin significantly increased the expression of Cyp7a1, confirming recent reports (*Byun et al., 2014*; *Fu, Cui & Klaassen, 2014*). The expression of Cyp7a1 is regulated by a mechanism involving the BA-activated farnesoid X receptor (FXR) in hepatocytes and enterocytes. Activation of FXR induces small heterodimer partner (SHP) to feedback inhibit Cyp7a1 gene transcription in liver (*Fu, Cui & Klaassen, 2014*). The present study shows that atorvastatin increased the expression of FXR and SHP similarly as Cyp7a1 (Fig. 4). The increased expression in SHP in the present study is unexpected compared to the literature (*Fu, Cui & Klaassen, 2014*). This discrepancy could be due to the long-term administration of atovarstatin (30 vs 7 days), and another mechanism might be involved.

Accumulation of bile acids in the liver is associated with alterations in bile acid transporters (*Fu, Cui & Klaassen, 2014*; *Klaassen & Aleksunes, 2010*; *Liu et al., 2013*). The retention and accumulation of hydrophobic bile acids inside hepatocytes during cholestasis have long been implicated as a major cause of BA-induced hepatic injury (*Perez & Briz, 2009*) and dysfunction of bile acid transporters. In the present study, atorvastatin decreased the expression of the main conjugated BA uptake transporter Ntcp and the main BA efflux transporter Bsep, but also decreased the expression of Ostα/β, which are BA transporters in the liver and ileum (Fig. 5). The decreased expression of bile acid transporters (Bsep, Ostα/β) might contribute to the accumulation of bile acids in the liver and stimulation of Cyp7a1 expression.

Circadian rhythms play important roles in liver metabolism and diseases including BA metabolism and Cyp7a1 (*Ferrell & Chiang, 2015a*). Cyp7a1 is a clock-driven gene showing a typical circadian rhythm (*Beltowski, Wojcicka & Jamroz-Wisniewska, 2009*; *Duez et al., 2008*; *Le Martelot et al., 2009*; *Noshiro et al., 2007*; *Zhang, Guo & Klaassen, 2011*). Circadian

variation of hepatic cholesterol synthesis is driven by HMG-CoA reductase (*Jurevics et al., 2000*), and HMG-CoA inhibitor atorvastatin could likely affect the circadian clock. The mammalian circadian clock is based on a transcription–translation feedback loop, which includes core genes: CLOCK:BMAL1, Npas2:Bmal1, and feedback genes Per, Cry and Nr1d1 (*Bozek et al., 2009*; *Ikeda & Nomura, 1997*; *Ye et al., 2014*). Some of the clock-controlled genes (Dbp, Dec2, and Nr1d1) are direct regulators required for the robust circadian rhythm of Cyp7a1 (*Noshiro et al., 2007*). Furthermore, SHP, the regulator of Cyp7a1, also has proved to be a downstream target of circadian clock (*Pan et al., 2010*). It has been observed that the dysregulation of Bmal1 and Npas2 in SHP-null mice (*Lee et al., 2015*; *Wu et al., 2016*). The present study shows that atorvastatin-induced increase in Cyp7a1 and SHP are associated with the core-clock dysregulation: Bmal1 and Npas2 were up-regulated after atorvastatin-treatment similar to induction of Cyp7a1. In contrast, Per1 and Per2, which provide negative feedback by interfering with the Npas/Bmal1/Clock transcriptional complexes, are down-regulated, and Cry1 was unchanged. In the bile duct ligation model of Per2-null mice Per2 played a protective effects of cholestasis, atorvastatin repressed the expression of this feedback genes may increase the risk of Liver injury (*Chen et al., 2013*). Thus, atorvastatin-induced disruption of hepatic BA homeostasis was related to the core-clock gene alterations.

Dbp, Nr1d1 and Tef are direct clock controlled output genes. Circadian expression of HMG-CoA was disrupted in Rev-Erba (Nr1d1)-null mice and the circadian rhythms of liver Cyp7a1, SHP were decreased in Nr1d1-null mice (*Beltowski, Wojcicka & Jamroz-Wisniewska, 2009*). In the present study, the expression of Nr1d1 tended to decrease but was not significant. The PAR-domain basic leucine zipper (PAR bZip) transcription factors, Dbp and Tef, accumulate in a highly circadian manner in liver participating in cellular metabolism (*Gachon et al., 2006*). In the present study, the expression of both Dbp and Tef was decreased by repeated administration of atorvastatin. Ethanol was show to alter the peripheral clock, including Dbp and Tef, without altering the central SCN clock (*Filiano et al., 2013*). Thus, alterations of liver circadian genes by atorvastatin could have implications in liver BA metabolism and disease (*Ferrell & Chiang, 2015a*).

Ideally, circadian rhythms would be better illustrated using multiple time points as shown in our recent publications with 6 time points (*Li et al., 2016*; *Lu et al., 2013*; *Xu et al., 2012*; *Zhang et al., 2012*). However, circadian rhythms can also be evaluated using a single time point as shown in our recent publications (*Li et al., 2017*). Thus, one time point evaluation is also informative, depending on the goals and sample feasibility of the study. In our preliminary study with 7-day atorvastatin administration, the upregulation of Cyp7a1 and Bmal1 at multiple time points was observed (Fig. S1).

In summary, repeated administration of atorvastatin altered bile acid metabolism and disposition. Atorvastatin increased the expression of the bile acid synthesis rate-limiting enzyme gene Cyp7a1, together with alterations in circadian clock genes. Chronopharmacology has emerged as a new insight into the therapeutic use of drugs (*Dallmann, Brown & Gachon, 2014*; *Ferrell & Chiang, 2015a*; *Li et al., 2016*), and the effects of atorvastatin on circadian clock is worthy of further investigation.

### Funding

This work was supported by the Chinese National Science Foundation grant 81160415. The funders had no role in study design, data collection and analysis, decision to publish, or preparation of the manuscript.

### Grant Disclosures

The following grant information was disclosed by the authors:
Chinese National Science Foundation: 81160415.

### Competing Interests

Jie Liu is an Academic Editor for PeerJ.

### Author Contributions

- Wen-Kai Li and Jie Liu conceived and designed the experiments, performed the experiments, analyzed the data, wrote the paper, prepared figures and/or tables, reviewed drafts of the paper.
- Huan Li performed the experiments, analyzed the data, reviewed drafts of the paper.
- Yuan-Fu Lu contributed reagents/materials/analysis tools, reviewed drafts of the paper.
- Ying-Ying Li performed the experiments, contributed reagents/materials/analysis tools.
- Zidong Donna Fu analyzed the data, wrote the paper, reviewed drafts of the paper.

### Animal Ethics

The following information was supplied relating to ethical approvals (i.e., approving body and any reference numbers):

All animal experiments were carried out in full compliance with the Guidance of Humane Care and Use of Laboratory Animals, and approved by the Animal Care and Use Committee of Zunyi Medical College (2013-5).

### Data Availability

The raw data has been supplied as a Supplementary File.

### Supplemental Information

Supplemental information for this article can be found online at http://dx.doi.org/10.7717/peerj.3348#supplemental-information.

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
