# Peer review of "Atorvastatin alters the expression of genes related to bile acid metabolism and circadian clock in livers of mice"

_PeerJ, doi:10.7717/peerj.3348_

## Round 0.1 · original submission · Major Revisions

As you can see from the review comments, your manuscript needs major revisions. Please take care to respond to all the points in detail.

Reviewer 1 ·

Basic reporting

The manuscript is well written and concise. However, there are several mistakes concerning grammar and interpunction (see below), I suggest the manuscript should be revised by a native speaker.

Experimental design

The authors conducted in vivo experiment, starting for blood biochemical test, histopathological observation, and ending by multiple gene expression such as Cyp7a1, Bmal1 and Npas2.

Validity of the findings

This study confirmed that atorvastatin increased the expression of the bile acid synthesis rate-limiting enzyme gene Cyp7a1, and found that this effect was associated with expression dysregulation of circadian clock genes.

Additional comments

The authors perform different experiments to explore the mechanism of cholestasis induced by atorvastatin.
To investigate the effect of atorvastatin on bile acid homeostasis, the authors observed that atorvastatin markedly increased the expression of the bile acid synthesis rate-limiting enzyme gene Cyp7a1, decreased the expression of bile acid transporters. Why not determine the level of bile acid in blood or in liver? The authors should explain the reason.

Minor comments:
1. Line 84: The number of mice for each group should be added.
2. Line 92: “mice” change to “the mice”; Line 97: “Blood” change to “The blood”.
3. Line 107 to 111: The interpunction should be checked and revised in an appropriate format.

Reviewer 2 ·

Basic reporting

The language in this article is OK, expect for some minor mistakes. For example: In the Rank 40, "cholestasis is the one of .....“ should be "cholestasis is one of...". in the rank 67, "Cyp7a1 is a circadian clock-driven gene having a typical circadian rhythm" may be "Cyp7a1 is a circadian clock-driven gene displaying a typical circadian rhythm"

Experimental design

Generally speaking, the circadian rhythm always describe the change during 24 hours, but not at some time point. in most of the research on the biorhythm, two time pionts at least should be included, such as 10AM and 10PM. In this article, the indice at only one time point is not very appropriate. And the experiment should be completed to observe the relavant indice at the same time point at night.

This study is to examine the effects of atorvastatin on the cyp7a1 and circadian clock genes, but the underlying mechanism is unclear. Both BA accumulation and circadian rhythm may play roles in regulating the expression of cyp7a1. So, the author should do some exprements to prove the irreplacebility of clock genes in the controlling the expression of cyp7a1.

Validity of the findings

The data in the excel is quite detailed, but I did not clear the caculation method of the QPCR. For example, in the excel of “PCR-BSEP”, dct=D2-16.38. What is 16.38? please explain it. And there is no duplication for each sample.

Additional comments

This study is focused on the cholestasis induced by statin, and the researchers examine the possible reasons of cholestasis from two perspective, BA sybthesis and transporting. So this study is meaningful and provide some evidence for understanding the adverse effect of statin. however, there are still some important comments.
1. How did the author choose the dose of the statin? Is it equal to the dose of the patients? How to caculate? As we know, 20mg per day is the common dose in the patients.
2. The observation at only one time point is difficult to be thought as the alteration relavant to the circadian rhythm. So, more expriments at other time point should be supplied.
2. In the Rank109, "reverse transcribe" should be "reversely transcribed".
3. Why were many index in the 100mg statin group lower than that of 50mg group? there is no obvious dose-dependent effect. Please give some explaination.
4. Please explain the underlying mechanism of the effect of statin on the clock genes. Statin plays different role on the different clock genes, some is positive and some is negative, please give some interpretation.
5. Why the clock targeted genes Dbp and Tef was decreased but the Cyp7a1 was increased? All these genes are under the control of circadian rhythm.

·

Basic reporting

1. It is noted that manuscript needs careful editing by someone with expertise in technical English editing paying particular attention to English grammar, spelling, and sentence structure so that the goals and results of the study are clear to the reader。
2. the backgroun is not sufficient introduced.
3. the figures' image quality should be improved.
4

Experimental design

1. the reserach is within the aim and scope of the journal.
2. the reserach quesstion are well defined.
3. the method and the infromation of replicate are not sufficient.

Validity of the findings

1. the novelty is not sufficient
2.the speculation is not well done.

Additional comments

In this manuscript, authors study the effects of atorvastatin on hepatic gene expression related to bile acid metabolism and homeostasis, however, there are many changes needed to be addressed, and this manuscript could be accpeted after major revision.
Weakesses are noted as listed below:
1. It is noted that manuscript needs careful editing by someone with expertise in technical English editing paying particular attention to English grammar, spelling, and sentence structure so that the goals and results of the study are clear to the reader.
2. There are few explanationes of this study design.
3. Authors could use more experiment method to check the differnent level of the expression of genes related to bile acid metabolism and circadian clock in livers of mice.
4. Some figures should be improved image quality, and the figure of Histopathology could be added the scale.

---

## Round 0.2 · Minor Revisions

The manuscript needs some final minor revisions as per the reviewer's comments.

Reviewer 1 ·

Basic reporting

See below

Experimental design

See below

Validity of the findings

See below

Additional comments

The authors have addressed my questions in the first review. I believe the revised manuscript is acceptable for publication.

Reviewer 2 ·

Basic reporting

No comment.

Experimental design

1. I still think only one time point is not very proper. Just as the supplemental data the author provided, the difference between the two groups is significant only at two time points. Thus, if you choose different time points, you will get different results.
2. The cyp7a1 expression is elevated at 10 am in this study, but the data in the supplemental figure showed that its expression did not alter after the statin administration at 10 am. Please give some explanation.

Validity of the findings

no comment.

Additional comments

1. The cyp7a1 is indeed regulated by the circadian rhythm. However, the change of cyp7a1 in this study can be induced by the BA accumulation and circadian rhythm. Which is more important?
2. the supplemental data showed the peak of BMAL1 expression is at midnight, but some articles showed its peak in the morning(FASEB J. 2014 Jan; 28(1): 176–194. doi: 10.1096/fj.13-232629). Why?

---

## Round 0.3 · accepted · Accept

Dear Jie,

Thank you for your submission to PeerJ.

I am writing to inform you that your manuscript - Atorvastatin alters the expression of genes related to bile acid metabolism and circadian clock in livers of mice - has been Accepted for publication. Congratulations!